# Evaluation of the Antigenotoxic Potential of Two Types of Chayote (*Sechium edule*) Juices

**DOI:** 10.3390/plants13152132

**Published:** 2024-08-01

**Authors:** Eduardo Madrigal-Santillán, Jacqueline Portillo-Reyes, José A. Morales-González, Luis F. Garcia-Melo, Estrella Serra-Pérez, Kristijan Vidović, Manuel Sánchez-Gutiérrez, Isela Álvarez-González, Eduardo Madrigal-Bujaidar

**Affiliations:** 1Unidad Casco de Santo Tomas, Instituto Politécnico Nacional, Escuela Superior de Medicina, Ciudad de Mexico 11340, Mexico; jacke_star230990@hotmail.com (J.P.-R.); jmorales101@yahoo.com.mx (J.A.M.-G.); 2Laboratorio de Nanotecnología e Ingeniería Molecular, Universidad Autónoma Metropolitana-Iztapalapa, Ciudad de Mexico 09340, Mexico; electronicfer@hotmail.com; 3Chemical Engineering and Materials Department, Faculty of Chemistry, Complutense University, 28040 Madrid, Spain; estrellaserra@ucm.es; 4National Institute of Chemistry, Department of Analytical Chemistry, Hajdrihova 19, 1000 Ljubljana, Slovenia; kristijan.vidovic@ki.si; 5Instituto de Ciencias de la Salud, Universidad Autónoma del Estado de Hidalgo, Pachuca de Soto 42080, Mexico; manuel_sanchez@uaeh.edu.mx; 6Escuela Nacional de Ciencias Biológicas, Instituto Politécnico Nacional, Unidad Profesional A. López Mateos, Ciudad de Mexico 07738, Mexico; isela.alvarez@gmail.com (I.Á.-G.); edumadrigal.bujaidar@gmail.com (E.M.-B.)

**Keywords:** antioxidant capacity, micronucleus assay, *sechium edule*

## Abstract

*Sechium edule* (Jacq.) Swartz is a perennial herbaceous climbing plant with tendrils and tuberous roots belonging to the Cucurbitaceae family. Its fruits (“chayote”), stems, roots, and leaves are edible and are commonly ingested by humans. It has shown medicinal properties attributed to its bioactive compounds (vitamins, phenolic acids, flavonoids, carotenoids, triterpenoids, polyphenolic compounds, phytosterols, and cucurbitacins), which together have been associated with the control and prevention of chronic and infectious diseases, highlighting its antibacterial, anti-cardiovascular/antihypertensive, antiepileptic, anti-inflammatory, hepatoprotective, antiproliferative, and antioxidant activities. The objective of the study was to determine the antigenotoxic potential of two types of fresh chayote juice (filtered (FChJ) and unfiltered (UFChJ)) against DNA damage produced by benzo[a]pyrene (B[a]P) using an in vivo mouse peripheral blood micronucleus assay (MN). The juices were consumed freely for 2 weeks. A negative control, a control group of each juice, a positive batch [B[a]P], and two combined batches (B[a]P plus FChJ or UFChJ) were included. Blood smears were stained and observed under a microscope to quantify the number of micronucleated normochromic erythrocytes (MNNEs). The results indicate: (a) B[a]P increased the frequency of MNNEs and reduced the rate of PEs; and (b) no juice produced toxic effects or induced MN. On the contrary, both juices were genoprotective. However, the most significant effect was presented by UFChJ at the end of the experiment (70%). It is suggested that UFChJ has a greater amount of fiber and/or phytochemicals that favor the therapeutic effect. Possibly, the genoprotection is also related to its antioxidant capacity.

## 1. Introduction

*Sechium edule* (Jacq.) Swartz is a perennial herbaceous climbing plant with tendrils and tuberous roots that belongs to the Cucurbitaceae [1,2] family. To date, 10 species have been characterized: 8 are wild (*S. chinantlense*, *S. compositum*, *S. hintonii*, *S. talamancense*, *S. panamense*, *S. pittieri*, *S. venosum,* and *S. vilosum*), and 2 are cultivated (*S. tobacco* and *S. edule*) [1]. Archaeological evidence suggests that it is native to Mexico and has been cultivated since pre-Columbian times [2,3]. Currently, it is cultivated in different tropical and subtropical areas of the world, and the main producing and exporting countries are Mexico, Costa Rica, Brazil, and the Dominican Republic [1,4]. Its fruits, popularly known as chayotes, are the main edible fleshy organ; they are pear-shaped and grow individually or in pairs on a shared peduncle. They are highly perishable, and unlike other cucurbits (pumpkin, cucumber, melon, and bitter apple), they contain a single seed, compressed and smooth, which can germinate again after 30 days of a harvest [1,2,4]. In addition to the fruits, their stems, tender leaves (known as *quelites*), and tuberous roots (called in Mexico *camochayote* and/or *chinchayote*) are considered an important component of the human diet, conventionally steamed, cooked, or sautéed, as well as fresh/raw in salads and in traditional stews with cream, curry, and/or pickled [4,5]. Despite the existence of different species, the *S. edule* variety “smooth green” is considered the fourth most consumed imported product, after avocado, tomato, and coffee, mainly attributed to its nutritional and biofunctional properties [1,2,3,4,5]. The nutritional composition of chayote is influenced by climate, region, growing conditions, plant age, and processing methods [1,2]. In general, it is rich in carbohydrates (soluble sugars, reducing sugars, and non-reducing sugars), starch, fiber (hemicelluloses, cellulose, lignin, and pectin), fatty acids (non-polar lipids, glycerolipids, and phospholipids), proteins, amino acids, vitamins (C, A, E, K, thiamin, riboflavin, niacin, pantothenic acid, pyryidoxin, and folic acid), and minerals (sodium, calcium, iron, magnesium, phosphorus, potassium, zinc, copper, and manganese) [1,2,3,4]. Both the fruit and stems, as well as the leaves and roots, have shown medicinal properties attributed to the plant’s various bioactive compounds (vitamins, polysaccharides, peroxidases, alkaloids, saponins, phenolic acids, flavonoids, carotenoids, triterpenoids, polyphenolic compounds, phytosterols, and cucurbitacins) [1,2,6], which together have been associated with the control, progression, and prevention of chronic and infectious diseases. Studies have highlighted the plant’s antibacterial [1,7,8], anti-cardiovascular/antihypertensive [1,9,10,11,12,13,14], anti-epileptic [1,15], anti-inflammatory [1,16,17], anti-ulcer [1,18], hepatoprotective [1,9,11,12,13,14,15,16,17,18,19], anticancer, antineoplastic, antiproliferative [20,21], and antioxidant [1,9,17,22,23,24,25,26,27] properties, though few studies have explored the antimutagenic, anticancer, and/or antigenotoxic effects of *Sechium edule* (Se). Furthermore, most studies were carried out in bacteria and/or in vitro cell culture models [20,27,28,29]. In this sense, Yen et al. (2001) observed that aqueous extracts of Se showed a greater inhibitory effect on mutagenicity induced by 2-amino-3-methyl-imidazo [4,5-f] quinoline (IQ) than against benzo[a]pyrene (B[a]P) and 4-nitroquinoline-N-oxide (NQNO) in *S. typhimurium* strains TA98 and TA100. Furthermore, they observed that the antimutagenic activity was partially reduced by heating them at 100 °C for 20 min, which suggests that the antimutagenic compounds present in the extracts could be altered by heat [20]. On the other hand, another ethanolic extract (EtOH) of fruits showed antiproliferative and cytotoxic activity on L-929 and HeLa tumor cell lines. Analysis by hydrogen nuclear magnetic resonance and gas chromatography coupled to mass spectrometry revealed that saturated fatty acid esters were the main components of the most active fraction [27]. Aguiñiga et al. (2015) observed that methanolic extract (MeOH) had a positive effect in the treatment of acute myeloid leukemia, specifically by inhibiting the proliferation of murine leukemic cell lines P388 (macrophages), J774 (monocytic) cell lines, and WEHI-3 (myelomonocytic). Their results also indicated a reduction in cell viability, an induction in the production of apoptotic bodies, and DNA fragmentation [28]. Finally, using two types of chromatography (thin layer and column), terpenes and flavonoids were identified from a MeOH extract of the fruit of *S. edule* var. amarus silvestrys. Specifically, cucurbitacins (B, D, E, and I) were identified in the terpene fraction, while rutin, phlorizidin, myricetin, quercetin, naringenin, phloretin, apigenin, and galangin were obtained from the flavonoid fraction. Likewise, it was concluded that the extract inhibited the proliferation of HeLa P-388 cells [29]. Given that in recent decades, the consumption of the *S. edule* variety “smooth green” has gained high acceptance, this document focuses on this species consumed in Mexico and in different parts of the world. The purpose of the present study was to determine the antigenotoxic potential of two types of fresh chayote juice (filtered (FChJ) and unfiltered (UFChJ)) against DNA damage produced by benzo[a]pyrene (B[a]P) using an in vivo mouse peripheral blood micronucleus assay (MN).

## 2. Results

Table 1 shows the results obtained from the weight of the animals. A normal weight increase was observed in the control animals, starting the second day. This increase continued in the following days. At the end of the experimental period, there was an increase of 15.06 g (a final average weight of 37.45 g). A similar behavior was found in animals treated with both types of juice, suggesting that FChJ and UFChJ did not produce toxicity. On the contrary, animals treated with B[a]P showed a greater increase in body weight compared to those in the control group from the third day of treatment. This increase was statistically significant during the fourth and ninth days, presenting an average weight difference of 2.73 g with respect to the control group. Finally, the animals treated with the combination of both juices plus B[a]P showed a protective effect during the experiment. In relation to the FChJ plus B[a]P group, there was a slight increase in their body weight, which was only significant on the eighth day. The best protective effect corresponded to the UFChJ animals. Both combined lots presented values comparable to the control at the end of the experimental period.

The quantity of food consumed by mice used in the experiment is shown in Table 2. The control mice as well as those administered with both types of chayote juice (FChJ or UFChJ) had similar food consumption throughout the experiment. Whereas, the B[a]P-treated mice showed a tendency to consume more food than the control animals. The same effect was observed in the animals treated with FChJ plus B[a]P. The rodents treated with the unfiltered juice and the mutagen showed a food intake similar to the control group. Regarding the amount of juice consumed during the experiment, all the rodents consumed the same amount of both juices (there were no significant differences).

The frequency of MNNEs in the studied groups is shown in Figure 1. Animals belonging to the control group and fresh chayote juices (FChJ and UFChJ) had no MN increase in the 14 days of the experiment; the mean value was 0.8 MN/2000 NE. The result suggests that both types of chayote juices were not genotoxic. On the other hand, the mice treated with B[a]P (250 mg/kg) manifested a significant increase since the fourth day of the assay, with the highest genotoxic damage of 8.6 MN/2000 NE at the end of the experiment. Regarding the effect of chayote juices on the MN rate induced by B[a]P, we detected moderate protection with FChJ. The highest protection (approximately 70%) was obtained with unfiltered chayote juice (UFChJ), where a significant decrease in MNNEs was observed from the sixth day of treatment until the end of the experiment.

Figure 2 shows the results obtained from the relation between PEs and the number of normochromatic erythrocytes (PE/NE index). Throughout the experimental period, the PE/EN index was similar in the control group and the animals treated with both chayote juices. There were no significant variations in the index. On the other hand, the mice administered with B[a]P showed a significant reduction in the rate of PEs during the last three days of the treatment period (70% with respect to the control group level). The result was similar to that observed in the animals treated with FChJ. However, the unfiltered chayote juice (UFChJ) produced an anticytotoxic effect that corresponded to a recovery of the PE/NE index of approximately 72% in relation to the animals administered with B[a]P.

## 3. Discussion

The World Health Organization (WHO) considers that traditional medicine/complementary and alternative medicine (TCAM) continues to be used in different populations around the world to treat and/or prevent the appearance and progression of chronic and infectious diseases, as it presents favorable factors that contribute to its increasing acceptance, such as easy access, diversity, relatively low cost, and most importantly, relatively low adverse toxic effects compared to allopathic medicine. Consequently, plants and their bioactive compounds (phytochemicals) have played an important role in human health care. Several typical species of the Cucurbitaceae family (melon “*Cucumis melo* L.”, cucumber “*Cucumis sativus* L.”, pumpkin “*Cucurbita moschata Duchesne*”, zucchini “*Cucurbita pepo* L.”, watermelon “*Citrullus lanatus*”, and chilacayote “*Cucurbita ficifolia*”) have horticultural importance and are included in TCAM. In particular, chayote (*S. edule*) has been associated with the control and prevention of some diseases attributed to the presence of its different phytochemicals (vitamins, polysaccharides, peroxidases, sterols, alkaloids, saponins, phenolic acids, flavonoids, carotenoids, triterpenoids, polyphenolic compounds, phytosterols, and cucurbitacins). In the particular case of its anticancer, antineoplastic, antiproliferative, and antimutagenic benefits, various authors agree that the combination of its bioactive compounds counteracts, reduces, and repairs the damage resulting from EOx and inflammation [6,17,20,21,26,30,31,32,33,34].

Based on this background, we evaluated the capacity of two types of fresh chayote juice (filtered (FChJ) and unfiltered (UFChJ)) to counteract the toxic effects of B[a]P, a compound belonging to the group of polycyclic aromatic hydrocarbons (PAHs). This chemical agent has been widely analyzed and studied for many years due to its high toxicity. Several authors using different types of assays have pointed out its mutagenic, carcinogenic, teratogenic, and clastogenic capacities [30,35,36,37,38,39]. However, regarding the effect that B[a]P produces on body weight, there are contradictory results. Some studies have indicated that when it is orally administered by inhalation or intraperitoneal injection at doses ranging from 50 to 2250 mg/kg, body weight could decrease in experiment animals [36,38,40,41,42]. There is also information showing the effects of smoking a large amount of tobacco, a major source of benzo[a]pyrene contamination (20–50 ng B[a]P per cigarette), which could also reduce body weight [43,44]. Likewise, studies carried out by Knukels et al. (2001) where body weight was quantified and the histopathology of some organs such as the liver, kidney, stomach, prostate, testicles, and ovaries was analyzed during the 30, 60, and 90 days after the administration of B[a]P in three doses (5, 50, and 100 mg/kg) by oral tube to F-344 rats (females and males) showed a significant reduction in body weight with the highest dose (100 mg/kg). Their results indicated that body weight can decrease and concluded that this effect is greater in males than in females [45]. In this sense, our study demonstrated the opposite, since there was an increase in food consumption, which possibly had an impact on the increase in body weight since the third day of treatment. This growth kinetics is similar to two previous studies, the first carried out by Irigaray et al. (2006), in which C57BI/6J mice were chronically intoxicated with a dose of 0.5 mg/kg B[a]P by intraperitoneal injection for 15 days. Their results reported a 40% increase in weight in comparison to the control group, without changes in their food intake (their evaluation parameter differed from ours). In addition, they discovered that by eliminating PAHs, the weight increase was not immediately corrected [46]. On the other hand, the second investigation, where some male CD-1 mice were orally treated for a week with the same PAH at a dose of 200 mg/kg (a dose similar to that reported in this document), confirmed a similar increase in body weight and consumption of food [30]. Although the mechanism(s) responsible for body weight control are not completely known, a possible explanation could be that a subchronic or chronic exposure to B[a]P induces changes in the adipose tissue metabolism and/or generates the formation of chemical interactions between the mutagen and the adipose cell. Probably, its high lipid solubility and long half-life favor its accumulation in this tissue, increasing the body mass index and causing fat accumulation by inhibiting lipolysis. This phenomenon is related to the blockage of adrenocorticotropic hormone (ACTH) receptors and beta-adrenergic receptors (β-1, -2, and -3) whose signals are coupled with the G protein system. Regarding the reason why the food intake was not modified, there is evidence that in “β-less” mice in which none of the three beta-adrenergic receptors are activated, there was an increase in body weight without any change in food intake [47,48,49].

The present study also demonstrated that none of the juices (FChJ or UFChJ) affected body weight or food intake, nor was it a genotoxic or cytotoxic agent. On the contrary, they reduced the toxicity and genotoxicity produced by PAHs. It is important to mention that in the literature, there are few in vivo studies where the same indicators or the same Se species are analyzed. Thus, the data obtained from other species of the Cucurbitaceae family, whose composition of bioactive compounds is similar, will be used and compared. It is worth mentioning that chayote is a traditional vegetable generally boiled for human consumption. Therefore, in order to avoid temperature alterations to its phytochemical and/or nutritional content, it was evaluated in juice form [20]. Melita-Rodríguez et al. (1984) were the first authors to administer Se juice to Wistar rats in order to evaluate its diuretic potential [50]. Another reason is the worldwide trend to increase the intake of different juices. In the case of chayote, as it does not contain fat and is rich in bioactive compounds that promote good health, chayote juice has been spread (*Virens levis* and *Nigrum spinosum* varieties) and combined with stevia (*Stevia rebaudiana* Bert.) and pineapple (*Ananas comosus*) to favor its neutral and tasteless flavor and promote its consumption [51]. Finally, Mandey et al. (2020) confirmed that cucumber (*Cucumis sativus*) seed juice improves the performance, body weight, and carcass parameters of broiler chickens [52]. Although *Virens levis*, *Nigrum spinosum,* and *Cucumis sativus* are different species from ours and their nutritional composition may vary, the combination of studies suggests favorable evidence regarding body weight. Therefore, we can suggest that this evaluation parameter may not be modified.

To analyze the antigenotoxic and anticytotoxic capacity of Se, it is worth recalling that B[a]P is rapidly absorbed by the intestine, and due to its highly lipophilic nature, it is transported in plasma through the lipoprotein system. Its distribution in different tissues favors accumulation in mammary glands and adipose tissue. It is metabolized through the cytochrome P450 system (specifically CYP1A1, CYP1A2, and CYP1B1) to reagents derived from dihydrodiol epoxide (B[a]P-7,8-dihydrodiol-10-epoxide or BPDE). These metabolites make a covalent bond to DNA, resulting in adducts that lead to mutations, uncontrolled cell growth, and therefore, the formation of tumors in different tissues (lung adenocarcinomas, lymphoproliferative tumors, hepatomas, and breast adenocarcinomas). Furthermore, with constant exposure to B[a]P, oxidative damage is induced in DNA, which plays an important role in the carcinogenic process due to quinone derivative production that easily generates reactive oxygen species (ROS) that are involved in its clastogenic capacity and the formation of chromosomal anomalies and DNA strand breaks [30,46,53,54].

The dose used in this experiment (250 mg/kg) once again confirms the genotoxic capacity of B[a]P, and together with the experiments carried out by other authors, we agree that this mutagen increases the frequency of micronuclei in reticulocytes and polychromatic erythrocytes (MNPEs) and/or normochromatic erythrocytes (MNNEs) when administered in doses ranging between 125 and 250 mg/kg [30,36,38,39,55]. Therefore, by comparing the results of our experiment with the aforementioned investigations, we can confirm that the MN technique was suitable to evaluate the genotoxic potential of B[a]P as well as the genoprotective capacity of Se [56]. Likewise, the evidence shown in the experimental conditions of the present study suggests that Se is not a toxic agent since it does not modify body weight, food intake, or genetic material. On the contrary, it shows an antigenotoxic capacity possibly attributed to the individual or combined effect of its phytochemicals, such as polysaccharides, vitamins, peroxidases, alkaloids, saponins, phenolic acids, flavonoids, carotenoids, triterpenoids, polyphenolic compounds, phytosterols, and cucurbitacins, many of which are known to have anticancer and/or antigenotoxic properties [20,21,26,29,33,34,57,58,59,60,61,62,63,64,65,66]. Previous studies carried out by Aguiñiga-Sánchez et al. (2017) and Da Cruz et al. (2022) coincide with the findings presented in this document by suggesting that the Se species is safe. In the first case, using CD-1 mice, the acute toxicity of a methanolic extract of the variety *Nigrum Spinosum* was evaluated, concluding that its LD50 was greater than 5000 mg/kg [63]. Afterwards, the efficacy and safety of a topical and oral administration of pumpkin seed oil (PSO) on the hair growth of BALB/c male mice for 7 days were determined. Both the comet assay and the micronucleus test confirmed that PSO did not induce genotoxic or mutagenic effects. On the contrary, it stimulates the proliferation of hair follicles without signs of any liver toxicity [67]. Recently, using a battery of genetic toxicity studies that included reverse mutagenicity and in vitro micronucleus assays (TK6 cell culture), the safety of a pectin extract enriched with rhamnogalacturonan-I (G3P-01) from pumpkin (*Cucurbita moschata* var. Dickinson) for use as an ingredient in foods and dietary supplements was determined. Their results again suggest that the pumpkin ingredient was well tolerated without any adverse effects on the clinical, macroscopic, hematological, blood chemistry, or histological pathology of the essential organs of the animals. Furthermore, G3P-01 was not genotoxic, and its dietary intake was safe up to concentrations of 36,000 ppm [68].

In relation to the antigenotoxic potential of fresh chayote juices (FChJ and UFChJ), one of the first pieces of scientific evidence that is consistent with our results is the isolation by repeated vacuum liquid chromatography of the compound SQFwB2D (24 alpha-ethyl-5 alpha-cholesta-7, trans-22-dien-3 beta-ol, or spinasterol) obtained from a chloroform extract of pumpkin flowers (*Cucurbita moschata Duchesne*). Using the in vivo micronucleus test, it was confirmed that SQFwB2D significantly reduced the mutagenicity of tetracycline (approximately 64%) at doses of 100 mg/kg [57]. Similarly, Taiwanese researchers analyzed the mutagenic and antimutagenic capacities of different plant parts (leaves, stems, and “immature and mature” fruits) from eight extracts of edible plants (including Se). Using the *S. typhimurium* test (strains TA98 and TA100), they demonstrated that the Se extract was not mutagenic; on the contrary, it strongly inhibited the mutagenicity of 2-amino-3-methyl-imidazo [4,5-f] quinoline (IQ) and B[a]P [20]. Elfiky et al. (2012) analyzed the genoprotective effect of PSO against the genotoxicity and cytotoxicity induced by Azathioprine or Imuran (an indirect acting agent that requires metabolic activation to develop its genotoxic and cytotoxic effects in both somatic and germ cells). Oral administration of PSO was effective in reducing the frequency of MNPEs and DNA fragmentation. In addition, the PE percentage increased and the PE/NE index improved. That is, PSO presented important antigenotoxic and anticytotoxic effects [59]. Finally, the pretreatment with an extract of *Citrullus colocynthis* L. fruits (at a dose of 200 mg/kg) significantly reduced the number of MNPEs generated by the oxidative damage of cyclophosphamide in mouse bone marrow cells. Again, mitotic activity was also favored by improving the PE/NE index [60].

Various authors consider the mechanisms of action of antigenotoxic agents to be complex. In general, they have been classified into extracellular action (inhibition of the uptake of mutagens and their endogenous formation, modification of the intestinal flora, formation of complexes and/or intestinal deactivation, favoring absorption of protective agents) and intracellular mechanisms (scavenging of reactive oxygen species, protection of DNA nucleophilic sites, stimulation of trapping and detoxification in non-target cells, modulation of xenobiotic metabolizing enzymes, activation of DNA repair mechanisms, regulation of signaling pathways, enhancement of apoptosis) [56,69,70,71,72,73]. In this sense, our findings are important in suggesting that the chemopreventive action of both types of fresh chayote juices (FChJ and UFChJ) can be established through different mechanisms. The first of them, and the most scientific evidence mentioned, is its antioxidant ability [17,21,26,29,33,39,51,59,60,61,62]. Another possibility is the decrease and/or presence of fiber (hemicellulose, cellulose, lignin, and pectin), which is a difference established between filtered and unfiltered juice. In this case, the UFChJ results suggest that the protective effect is related to the decrease in intestinal absorption of some mutagenic agents, including B[a]P [56,74,75,76]. However, we can suggest other alternatives considering that different plants or fruits (especially in the form of extracts) have been used as natural dietary supplements to counteract the cytotoxic effects resulting from exposure to mutagens and carcinogens through medications, diet, or the environment. The chemopreventive action of extracts has been related to their ability to improve the activities of carcinogen-metabolizing enzymes and/or bind with toxins, thus reducing their critical effective concentrations. For example, some flavonoids (bioactive compounds that are also present in the chemical composition of Se) have a certain activity on cytochrome P450-dependent enzymes, which contributes to their anticancer capacity [56,77,78].

## 4. Materials and Methods

### 4.1. Chemicals

The following compounds were purchased from Merck Chemicals (Mexico City, Mexico): benzo[a]pyrene (B[a]P), methanol, ethanol, monobasic potassium phosphate (KH_2_PO_4_), sodium phosphate dibasic (Na_2_HPO_4_), and Giemsa stain.

### 4.2. Animals

Male mice strain CD-1 were obtained from the Institute of Health Sciences at the Autonomous University of Hidalgo (Hidalgo, Mexico) and maintained according to the norms of the Institutional Ethics Animal Care and Use Committee (CIECUAL) under standard conditions with a 12 h/12 h light/dark cycle, constant temperature (22 ± 2 °C), and humidity (50 ± 10%). Food (Lab Diet^®^ (St Louis, MO, USA) 5013) and water (depending on the experimental group) were freely available in their home cages. The adjustment time period for the animals was one week before the treatments. The experimental procedures were approved by CIECUAL (Approval Number: CIECUAL-V-I/013/2023) of the Autonomous University of the State of Hidalgo (AUSH).

### 4.3. Plant Material and Obtaining and Preparation of Fresh Chayote Juices

In the study, fruits of the *S. edule* variety “smooth green” were used (Figure 3). The plant species was corroborated and taxonomically classified by professors of the biology school at the Institute of Basic Sciences and Engineering of the AUSH. The fresh chayotes were purchased at the local market 1° de Mayo (Pachuca, Hidalgo, Mexico), washed with distilled water, and cut into large slices, preserving the skin, pulp, and seed. The juice was obtained using a domestic juicer (T-fal Frutelia Plus ZE3708MX, Mexico City, Mexico). Subsequently, the juice was filtered (to remove most of the soluble fiber) using Whatman^®^ (Maidstone, UK) grade 5 cellulose filter paper (pore size: 2.5 µm). Finally, the filtered juice (FChJ) was diluted 50% with distilled water (*V*/*V*). The unfiltered chayote juice (UFChJ) was obtained in the same way without subjecting it to the Whatman^®^ grade 5 cellulose filter paper process. UFChJ was also diluted to 50% with distilled water (*V*/*V*).

### 4.4. Antigenotoxicity/Genotoxicity Protocol

The protocol was carried out according to the methodology described by Madrigal-Santillán et al. (2012) [30]. A total of 36 male mice strain CD-1 with a mean weight of 22 ± 2 g were organized in groups of six individuals each, as follows: a negative control group (without any treatment), control batches of fresh chayote juices (FChJ or UFChJ), a positive group treated with B[a]P (250 mg/kg body weight, dissolved in 200 μL corn oil), and two combined lots (B[a]P plus FChJ or UFChJ). B[a]P was administered orally by intragastric tube throughout the experimental period, according to the experimental groups, while FChJ and UFChJ were freely consumed by the animals. The same amount of food and the different fresh chayote juices (FChJ and UFChJ) were provided in each cage. The weight of each mouse was measured daily during the investigation period. The difference in weight per cage indicated the amount of juice and food consumed. A micronucleus assay was used to determine the genotoxic and antigenotoxic capacities of the compounds. The rate of micronucleated normochromatic erythrocytes (MNNEs) was determined before the experimental treatment and at 0, 48, 96, 144, 192, 240, 288, and 336 h after the treatment. We took two blood smears from the tail of each animal, fixed in methanol for 5 min, and stained for 20 min with a 4% Giemsa solution made in phosphate-buffered saline at a pH of 6.8. Two thousand erythrocytes per animal were scored to determine the rate of MNNEs. In order to evaluate the bone marrow cytotoxicity, we scored 2000 erythrocytes per animal and established the rate of polychromatic erythrocytes (PEs) with respect to the number of normochromatic erythrocytes (PE/NE index).

### 4.5. Data Analysis

All the data obtained were analyzed with the statistical program INSTAT version 3.0, and an ANOVA was performed to determine if the data among all groups were normally distributed. In that case, Tukey–Kramer tests as a post-test were performed to establish the statistical differences between groups in each assay. A *p*-value ≤ 0.05 was considered significant.

## 5. Perspectives and Conclusions

In the present research, we have shown that treatments with fresh chayote juices significantly prevent B[a]P-induced damage in mice, thus reducing the frequency of micronuclei normochromatic erythrocytes (MNNEs). This result increases the beneficial properties of Se mentioned in the introduction (antibacterial, anticardiovascular, antihypertensive, antiepileptic, anti-inflammatory, antiulcerative, hepatoprotective, and antiproliferative). In general, different authors, after carrying out phytochemical analyses of aqueous, ethanolic, and methanolic extracts of fruits from *S. edule var. Virens levis* (the chayote variety with the greatest export from Mexico and very similar to the one evaluated in this document), coincide and highlight the identification of flavonoids (mainly rutin, phlorizin, myricetin, phloretin, naringenin, and apigenin) and phenolic acids such as gallic, syringic, vanillic, p-hydroxybenzoic, caffeic, ferulic, p-coumaric, and chlorogenic; the latter being the one with the greatest presence. Cucurbitacins whose concentration varies between 0.11 and 1.45 mg/g of fresh weight predominate those of types D, I, E, and B. In addition, the high content of dietary fiber (raw fiber) in the fruits varies between 0.4% and 7.6% when they are raw and 2.2% when they are cooked [2,63,79,80]. Therefore, an important perspective of this document would be to carry out a phytochemical study of the juices (through high-performance liquid chromatography (HPLC), mass spectrometry (MS) analysis, and UV-Vis absorption spectrophotometry) to characterize their main active compounds. Likewise, the antigenotoxic analysis of its individually isolated bioactive compounds would be a relevant point of research.

In conclusion, our data also suggest that the presence of fiber (whose mechanism of extracellular action is related to its adsorbent capacity, inhibiting or reducing the absorption of carcinogens and/or eliminating them from the body) and the antioxidant capacity of juices may be involved in the plant’s genoprotective effect, predominantly fresh, unfiltered chayote juice (UFChJ).

## Figures and Tables

**Figure 1 plants-13-02132-f001:**
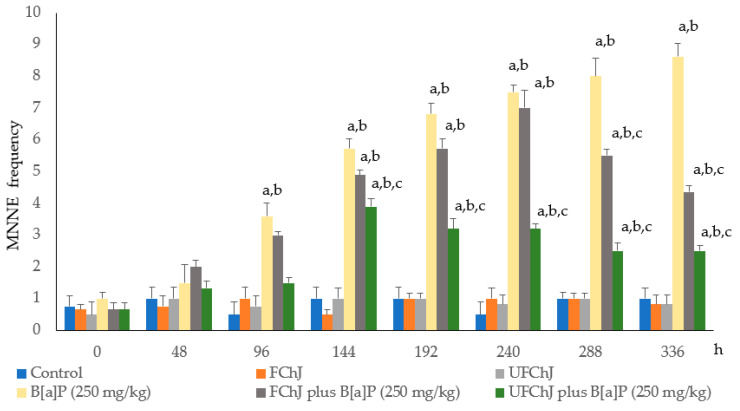
Frequency of normochromatic micronucleated erythrocytes (MNNEs) in mice treated with two types of fresh chayote juices (filtered juice (FChJ) or unfiltered juice (UFChJ)) and benzo[a]pyrene (X ± SD). Two thousand erythrocytes per animal stained with Giemsa were scored to determine the rate of MNNEs. Values represent the mean ± S.D. of six mice per group. The letters show significant statistical differences as follows: ^a^ with respect to the initial day, ^b^ with respect to the control value, and ^c^ with respect to the value in the group treated with B[a]P. Analysis of variance and Tukey–Kramer tests (*p* ≤ 0.05) were conducted.

**Figure 2 plants-13-02132-f002:**
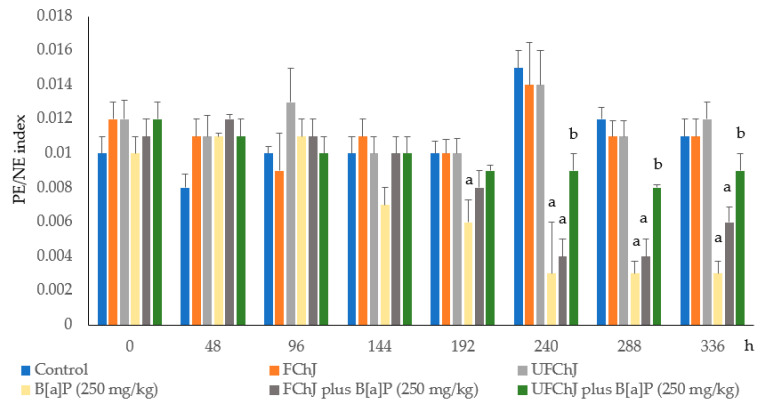
Relationship between the number of polycromatic erythrocytes with respect to the number of normochromatic erythrocytes (PE/NE index). Mice were treated with two types of fresh chayote juices (filtered juice (FChJ) or unfiltered juice (UFChJ)) and benzo[a]pyrene (X ± SD). The amount of each type of erythrocyte was determined at 2000 erythrocytes per animal using the Giemsa stain. Values represent the mean ± S.D. of six mice per group. The letters show significant statistical differences as follows: ^a^ with respect to the control value, and ^b^ with respect to the value in the group treated with B[a]P. Analysis of variance and Tukey–Kramer tests (*p* ≤ 0.05) were conducted.

**Figure 3 plants-13-02132-f003:**
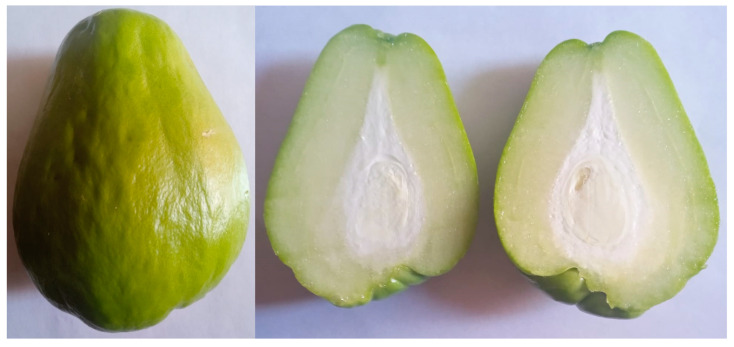
Fruits of the *S. edule* variety “smooth green”.

**Table 1 plants-13-02132-t001:** Weight gain (g) in mice treated with two types of fresh chayote juices (filtered juice (FChJ) or unfiltered juice (UFChJ)) and benzo[a]pyrene (X ± SD).

Day	Control	FChJ	UFChJ	B[a]P (250 mg/kg)	FChJ Plus B[a]P (250 mg/kg)	UFChJ Plus B[a]P (250 mg/kg)
0	22.39 ± 1.70	23.05 ± 1.66	21.75 ± 1.59	23.05 ± 0.66	22.75 ± 0.83	22.05 ± 1.71
1	23.15 ± 2.31	23.55 ± 2.31	22.39 ± 1.71	24.15 ± 2.32	23.55 ± 2.32	23.05 ± 2.32
2	24.05 ± 2.45 ^a^	24.75 ± 0.84	23.75 ± 0.83 ^a^	24.75 ± 2.59	24.75 ± 2.15 ^a^	24.15 ± 2.46 ^a^
3	24.75 ± 2.51 ^a^	25.85 ± 2.32 ^a^	24.55 ± 2.32 ^a^	26.55 ± 1.91 ^a,b^	25.25 ± 0.99 ^a^	25.15 ± 2.59 ^a^
4	25.15 ± 2.31 ^a^	26.10 ± 1.71 ^a^	25.05 ± 2.59 ^a^	27.92 ± 0.96 ^a,b^	26.05 ± 0.96 ^a,c^	25.75 ± 2.32 ^a,c^
5	26.05 ± 2.43 ^a^	26.75 ± 2.46 ^a^	26.15 ± 2.46 ^a^	28.20 ± 2.15 ^a,b^	27.05 ± 0.92 ^a^	26.75 ± 2.46 ^a^
6	27.12 ± 2.52 ^a^	27.32 ± 2.59 ^a^	27.25 ± 2.59 ^a^	29.15 ± 2.03 ^a,b^	28.10 ± 1.03 ^a^	28.12 ± 2.59 ^a^
7	28.02 ± 2.31 ^a^	28.75 ± 2.32 ^a^	28.22 ± 2.32 ^a^	31.73 ± 1.40 ^a,b^	29.05 ± 1.40 ^a,c^	28.75 ± 2.32 ^a,c^
8	29.21 ± 2.41 ^a^	30.05 ± 2.46 ^a^	29.15 ± 2.46 ^a^	33.33 ± 2.12 ^a,b^	31.91 ± 0.62 ^a,b^	29.75 ± 2.46 ^a,c^
9	31.05 ± 2.52 ^a^	31.25 ± 2.59 ^a^	31.55 ± 2.59 ^a^	34.10 ± 1.94 ^a,b^	32.75 ± 1.37 ^a,b^	30.75 ± 2.59 ^a,c^
10	32.05 ± 2.31 ^a^	32.55 ± 2.32 ^a^	32.75 ± 2.32 ^a^	33.69 ± 1.74 ^a^	32.75 ± 1.51 ^a^	33.05 ± 2.32 ^a^
11	33.05 ± 2.42 ^a^	33.75 ± 2.46 ^a^	33.21 ± 2.46 ^a^	34.02 ± 1.21 ^a^	34.02 ± 1.21 ^a^	34.05 ± 2.46 ^a^
12	34.29 ± 2.55 ^a^	34.12 ± 2.59 ^a^	35.75 ± 2.59 ^a^	35.12 ± 0.83 ^a^	34.75 ± 0.26 ^a^	34.75 ± 2.59 ^a^
13	36.75 ± 2.31 ^a^	36.55 ± 2.32 ^a^	36.05 ± 2.32 ^a^	36.35 ± 1.16 ^a^	36.12 ± 1.16 ^a^	36.05 ± 2.32 ^a^
14	37.45 ± 2.42 ^a^	37.05 ± 2.46 ^a^	37.75 ± 2.46 ^a^	38.05 ± 1.46 ^a^	37.15 ± 2.46 ^a^	37.05 ± 2.46 ^a^

The weight of each mouse was determined daily during the entire experiment. The data are average values for six animals/group. The letters show significant statistical differences as follows: ^a^ with respect to the initial day, ^b^ with respect to the control value, and ^c^ with respect to the value in the group treated with B[a]P. Analysis of variance and Tukey–Kramer tests (*p* ≤ 0.05) were conducted.

**Table 2 plants-13-02132-t002:** Food consumption (g) in mice treated with two types of fresh chayote juices (filtered juice (FChJ) or unfiltered juice (UFChJ)) and benzo[a]pyrene (X ± SD).

Day	Control	FChJ	UFChJ	B[a]P (250 mg/kg)	FChJ Plus B[a]P (250 mg/kg)	UFChJ Plus B[a]P (250 mg/kg)
0	2.91 ± 0.08	3.00 ± 0.0	2.85 ± 0.08	2.97 ± 0.12	3.05 ± 0.0	3.03 ± 0.01
1	2.98 ± 0.12	3.00 ± 0.07	3.05 ± 0.26	3.08 ± 0.36	3.10 ± 0.02	3.09 ± 0.08
2	3.03 ± 0.09	3.03 ± 0.04	3.06 ± 0.22	3.16 ± 0.09	3.17 ± 0.07	3.13 ± 0.14
3	3.13 ± 0.12	3.10 ± 0.07	3.19 ± 0.26	3.17 ± 0.11	3.19 ± 0.01	3.18 ± 0.17
4	3.25 ± 0.12	3.16 ± 0.07	3.20 ± 0.26	3.22 ± 0.11	3.19 ± 0.01	3.20 ± 0.26
5	3.35 ± 0.12	3.33 ± 0.07	3.30 ± 0.26	3.32 ± 0.11	3.37 ± 0.01	3.35 ± 0.17
6	3.50 ± 0.09	3.48 ± 0.26	3.89 ± 0.22	3.66 ± 0.09	3.44 ± 0.02	3.47 ± 0.04
7	3.87 ± 0.23	3.60 ± 0.08	4.06 ± 0.41	3.89 ± 0.05	3.72 ± 0.02	3.55 ± 0.04
8	4.02 ± 0.30	3.95 ± 0.26	4.00 ± 0.22	4.62 ± 0.10 ^a,b^	4.78 ± 0.01 ^a,b^	3.90 ± 0.07
9	4.07 ± 0.30	3.98 ± 0.26	4.00 ± 0.26	4.89 ± 0.30 ^a,b^	4.60 ± 0.26 ^a^	4.00 ± 0.17 ^c^
10	4.00 ± 0.10	4.02 ± 0.04	4.06 ± 0.36	4.95 ± 0.14 ^a,b^	4.73 ± 0.22 ^a,b^	4.05 ± 0.05 ^c^
11	4.06 ± 0.23	4.04 ± 0.23	4.09 ± 0.24	4.99 ± 0.17 ^a,b^	4.04 ± 0.11 ^c^	4.02 ± 0.07 ^c^
12	4.00 ± 0.07	4.06 ± 0.01	4.05 ± 0.14	4.75 ± 0.04 ^a,b^	4.02 ± 0.07 ^c^	3.99 ± 0.12 ^c^
13	3.99 ± 0.11	3.97 ± 0.10	3.98 ± 0.07	4.83 ± 0.07 ^a,b^	3.98 ± 0.04 ^c^	4.03 ± 0.01 ^c^
14	4.02 ± 0.09	4.08 ± 0.02	4.00 ± 0.02	4.95 ± 0.05 ^a,b^	4.00 ± 0.01 ^c^	4.06 ± 0.02 ^c^

The amount of food ingested per cage was determined daily by obtaining the difference in weight before and after each measurement. The data are average values for six animals/group. The letters show significant statistical differences as follows: ^a^ with respect to the initial day, ^b^ with respect to the control value, and ^c^ with respect to the value in the group treated with B[a]P. Analysis of variance and Tukey–Kramer tests (*p* ≤ 0.05) were conducted.

## Data Availability

Data are contained within the article.

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
