# Peer review of "Evaluation of the Antigenotoxic Potential of Two Types of Chayote (Sechium edule) Juices"

_plants, 2024, doi:10.3390/plants13152132_

Round 1
Reviewer 1 Report
Comments and Suggestions for Authors
Dear Authors,
I read your manuscript and I want to make just two observations:
Line 106-113 - it is not very clear for me, how the weight is increased in the experimental group. Even if the weight evolution is described by Table 1, the explanation does not convince me. Can you reformulate it, please?
Line 411-412 - you state "Our data also suggest that the presence of fiber and the antioxidant capacity may be involved in its genoprotective effect." Can you explain better how the fiber influence the genoprotective effect? And I consider that a conclusion regarding which juice has the better effect should be appropriate in the Conclusion section.
Best regards!
Comments on the Quality of English LanguageManuscript has to be corrected for a few typo errors.
Author Response
Dear reviewer
The authors appreciate the comments and observations of the article
We have considered all suggestions and observations
Please check the attached file
Thanks for everything
Receive a cordial greeting
- Page 3. Lines 115-123. The suggestion was considered. The paragraph corresponding to the results in Table 1 was rewritten and/or modified.
- Page 12. Lines 447-450. The suggestion was considered. A paragraph was included where the mechanism of action of the fiber is briefly mentioned. Also a conclusion indicating that the most significant effect corresponded to unfiltered chayote juice.

Reviewer 2 Report
Comments and Suggestions for Authors
in my opinion, the study is current and interesting.
however, I would suggest presenting the results in a more attractive and easier to view form.
the tables are quite difficult to follow and I would suggest the form of a graph of the most relevant results.
I would also suggest a diagram of the article, an graphical abstract for example or an easy-to-understand and follow diagram of the method implemented.
overall, the article is well structured and contributes important data to the current medical literature.
Author Response
Dear reviewer
The authors appreciate the comments and observations of the article
We have considered all suggestions and observations
Please check the attached file
Thanks for everything
Receive a cordial greeting

Reviewer 3 Report
Comments and Suggestions for Authors
Title: Determination of the antigenotoxic potential of chayote (Sechium edule).
The title of the manuscript is consistent with the topic of the study, but maybe it's worth mentioning in the title that was tested of two types of chayote juices (filtered and unfiltered). The main topic of this article was to determine the antigenotoxic potential of two types of fresh chayote juice [filtered (FChJ) and unfiltered (UFChJ)] against DNA damage produced by benzo[a]pyrene (B[a]P) using an in vivo mouse peripheral blood micronucleus assay (MN). In the investigations the Authors have demonstrated that the treatment with fresh chayote juices (FChJ and UFChJ) significantly prevents the damage induced by B[a]P in mice, thereby reducing the frequency of micronuclei. Presented data also suggest that the presence of fiber and the antioxidant capacity may be involved in its genoprotective effect.
The work requires refinement and contains numerous errors, both linguistic and factual. In the section of the discussion, the authors draw constructive conclusions. The scope of literature data is up-to-date and consistent with the subject of the research undertaken.
Comments and suggestions for Authors:
· I would add information to the title of the manuscript that two types of juices have been tested, filtered and unfiltered.
· Citations are used inconsistently, for example [9-14], [22,23,24,25].
· The text should be written more carefully, for example, dashes in the middle of words, dots in the middle of sentences, etc.
· In my opinion, the publication lacks of own phytochemical studies, especially since the keywords of the article include "phytochemicals". It would be good if both juices were subjected to a chromatography process. If the main active compounds were characterized (identification MS, UV-VIS), the research would be more complete.
· It would also be interesting to quantify the active compounds in these two types of chayote juices.
In my opinion, the manuscript can be released for publication only after corrections are made.

Author Response

(The authors gave the same response as above.)

Round 2
Reviewer 1 Report
Comments and Suggestions for Authors
Dear Author,
I appreciate your availability in changing the mentioned issues in my previous report.
I agree the publication in the present form.
Wish you all the best!
Reviewer 2 Report
Comments and Suggestions for Authors
Dear authors, your article looks much better now and I think it is worthy of being published.
All the best!
Reviewer 3 Report
Comments and Suggestions for Authors
The Authors add all coreccts. The Manuscript is very good finish version.